# Computer-Assisted Design of Environmentally Friendly and Light-Stable Fluorescent Dyes for Textile Applications

**DOI:** 10.3390/ijms20235971

**Published:** 2019-11-27

**Authors:** Songsong Tang, Guoqiang Chen, Gang Sun

**Affiliations:** 1National Engineering Laboratory for Modern Silk, College of Textile and Clothing Engineering, Soochow University, Suzhou 215123, China; songsongtang@outlook.com; 2Division of Textiles and Clothing, University of California, Davis, CA 95616, USA

**Keywords:** fluorescence, hemicyanine, photo-stability, Gaussian calculations

## Abstract

Five potentially environmentally friendly and light-stable hemicyanine dyes were designed based on integrated consideration of photo, environmental, and computational chemistry as well as textile applications. Two of them were synthesized and applied in dyeing polyacrylonitrile (PAN), cotton, and nylon fabrics, and demonstrated the desired properties speculated by the programs. The computer-assisted analytical processes includes estimation of the maximum absorption and emission wavelengths, aquatic environmental toxicity, affinity to fibers, and photo-stability. This procedure could effectively narrow down discovery of new potential dye structures, greatly reduce and prevent complex and expensive preparation processes, and significantly improve the development efficiency of novel environmentally friendly dyes.

## 1. Introduction

Fluorescent dyes, an important kind of functional dyes, have been used in many fields such as solar cells [1,2], biological imaging [3,4], probes [5,6], and optical nanoscopy [7,8]. However, many fluorescent dyes have been banned and will be phased out in commercial uses due to the concerns of human and environmental health as well as increasing regulations. For example, many azo dyes are prohibited to be used in textiles in European countries since aromatic amines, as potential metabolized products, are harmful to humans and the environment [9,10].

Hemicyanine fluorescent dyes usually have high molar absorption coefficients, broad spectrum width of absorption, and high-quantum yields, because of good coplanarity and conjugated systems with suitable electron donating (EDG) and electron withdrawing groups (EWG) [11,12,13,14]. However, these dyes may not have good light stability, especially when applying them in textile materials. Taking compound Z1 (Figure 1a,b) as an example, it possesses perfect chemical structural features and great fluorescence properties (Figure 1c), emitting yellow–red fluorescence [15,16,17,18], but the light reflectivity of polyacrylonitrile (PAN) fabric dyed by the compound Z1 decreased greatly when it was exposed to the light for 5 h (see Figure 1d). Thus, it was hardly used in dyeing fabrics. Fortunately, there are some different ways to improve the photo-stability of the dyes, for example improving the photo-stability of dyes using TiO_2_ [19], while computational chemistry could estimate behaviors of dyes under light according to quantum chemistry, which could provide theoretical speculations of photo-chemical/light fastness of the dyes [20,21,22,23,24,25].

Structural design of fluorescent dyes could be conducted by using computational programs, which could avoid unnecessary work in searching for dyes with improved light-stability. Potential environmental impacts of any designed dye could be estimated by using an ecological structure activity relationships (ECOSAR) predictive model, which is maintained by US Environmental Protection Agency (US-EPA) for predicting aquatic toxicity of chemicals [26,27], while suitability of the designed dyes on different fibers can be estimated by using a Hansen Solubility parameter theory (HSP) [27,28,29,30]. In this manuscript, five hemicyanine dyes were designed by using similar starting chemicals and reactions as environmentally friendly fluorescent dyes with improved photo-stability for textile applications. Gaussian 09, a computational program, was employed in analysis of chemical structures of the dyes, while the ECOSAR was adopted to estimate the aquatic-toxicity of raw materials, intermediates, and the dyes. Furthermore, the photo-stability of the dyes were analyzed as well after the affinities of the dyes to fibers were estimated by using Hansen solubility parameters (HSP). In addition, two of the dyes with expected good fluorescent features were prepared, and basic properties of the dyed materials were tested.

## 2. Results and Discussion

### 2.1. Biosafety Analysis of Synthetic Processes

With the results of analysis on the synthesis of the proposed dye molecules, the potential toxicities of the raw materials, all intermediates, and the dyes were estimated by using the ECOSAR. Among all raw materials, compounds A2, A3, A4, and A5 showed LC_50_ and EC_50_ values higher than 1000, and especially the LC_50_ (fish, 96h) and LC_50_ (daphnid, 48h) values of compounds A2, A3, and A4 are greater than 10 million. It means that the starting compounds A2, A3, A4, and A5 are more environmentally friendly than the compounds A1 and A6.

Compounds A1, A2, and A5 have similar structures but very different toxicity values due to the different substituents. The parent chemical (A1) has LC_50_ (fish, 96 h), LC_50_ (daphnid, 48 h), and EC_50_ (green algae, 96 h) values of 292.3, 156.33, and 90.92, respectively. Thus, the A1 is toxic to aquatic lives—especially to the green algae. The corresponding values of the A2 are about 1000 times greater than those of the A1, while the values of LC_50_ and EC_50_ of the A5 are about 20–38 times larger than those of the A1. The results reveal that both sulfonic acid (–SO_3_H) and carboxyl (–COOH) groups could increase LC_50_ and EC_50_ values and decrease the toxicity of the chemicals, while the sulfonic acid group is more effective than carboxyl group to decrease the toxicity of chemicals. The positions of different substituents on the pyridinium ring have different influence on their toxicities. The carboxyl group at the 3 position (compound A5) of the ring increased the values of LC_50_ and EC_50_ of the chemicals, but the carboxyl group at the 2 position (compound A6) increases the toxicity of the chemical. Different to the carboxyl group, the sulfonic acid group (see compounds A2, A3, and A4 in Appendix A and Table 1) decreases the toxicity of the chemicals on any positions.

All pyridinium salts (compounds B1 to B6) are less toxic than the starting pyridine derivatives based on their higher values of LC_50_ and EC_50_. However, the toxicities of the designed dyes, except Z6, are relatively higher than the intermediates; they are significantly lower than the starting compounds and Z1. The increased values of LC_50_ and EC_50_ of chemicals are possibly due to the increased water solubility.

### 2.2. Fluorescence Properties of Designed Dyes

Fluorescent properties of these designed dyes were speculated by using the computational program as well. The dye Z1 (see Appendix A) still possess coplanar conjugated structures even though different substituent groups are not in the same plane. It would emit fluorescence after excited by relevant light.

Appendix A shows the emission and absorption data, calculated by the Gaussian 09 based on the b3lyp/6-31G(d) level of N-methylacridinium chloride in water. The calculated maximum emission wavelength was 482.73 nm (2.5648 eV), which was close to the data (498 nm) in reference [31], while the maximum absorption wavelength was 412.93 nm—well matched with the reference data (415 nm) [31]. The maximum absorption and emission wavelengths of designed dyes (shown in Table 2) are varied from that of Z1 but still can emit visible lights after excited by relevant light.

### 2.3. Dyeing Properties Analysis of Designed Dyes

The affinity between the dyes and fibers could be analyzed by using Hansen solubility parameter theory (HSP), which has been employed in estimating affinity and interactions between molecules [32,33]. HSP distances (Ra values) between the different molecules in Hansen space can be calculated by Equation (1) [34,35]. The smaller Ra values represent better affinity between two molecules.
(1)$$\mathrm{Ra}=\sqrt{4{\Delta \delta }_{D}^{2}+{\Delta \delta }_{P}^{2}+{\Delta \delta }_{H}^{2}}$$

δ_D_: The energy from dispersion forces between molecules.

δ_P_: The energy from dipolar forces between molecules.

δ_H_: The energy from hydrogen bonds.

If a dye has high affinity to a fiber, it will be more attractive to the fiber [36], showing better dyeing properties. Five popular natural and chemical fibers were selected to estimate the affinities to the designed dyes. CI Disperse Yellow 11, one of the earliest commercial fluorescent dyes and widely employed in coloration of synthetic fibers (i.e., nylon and PET) and plastics [37,38], was selected randomly to show the practicality of this calculation model.

As shown in Table 3, Ra1 to Ra6 values reflect affinities of a dye to water, cellulose, acrylics (PAN), polyester (PET), Nylon 6, and Nylon 66, respectively. Ra4, Ra5, and Ra6 of CI Disperse Yellow 11 are 8.27, 7.08, and 7.16, which are much lower than Ra1 (36.49) and Ra2 (16.64), indicating that the CI Disperse Yellow has better affinity with PET, Nylon 6, and Nylon 66 than water and cellulose—consistent with the existing data. The designed dye Z1 possess low but close Ra3, Ra4, Ra5, and Ra6 values, which are lower than Ra2 (23.69) and Ra1 (43.02). It means the Z1 could dye PAN, PET, Nylon 6, and Nylon 66 with potential good wash fastness. The Z1 could color cellulose, however due to the small difference of Ra2 and Ra1 values, the wash fastness of the Z1 on cellulose may not be so good. Based on the HSP results, the dye Z2 should have the same dyeing properties to the dyes Z3 and Z4, while the dye Z5 should have same dyeing properties as the dye Z6. Similarly, the dyes Z2, Z3, and Z4 could dye all five fibers but may have different color fastness on the products.

Comparing the dyes Z1–Z6, they have similar conjugated structures but different dyeing properties due to varied substitutes. Both of the carboxyl and sulfonic acid groups can improve the affinity between dyes and certain polymers, but the sulfonic acid group showed stronger effect than that of the carboxyl group. It should be pointed that the disperse dyes, such like CI Disperse Yellow 11, were hardly used to color PAN fiber, while some water-soluble dyes are difficult to dye PET fiber in practice. Thus, we should analyze the dyeing properties not only based on the HSP distance, but also need to consider the practical/commercial knowledge of dyeing fibers.

### 2.4. Photo-Stability Analysis of Designed Dyes

Photo-stability, an important parameter of dyes, is the main concern when selecting dyes for textile application. According to the literature [20,24], photo-degradation of hemicyanine dyes could be caused by reactions between the dyes and reactive oxygen species-such as singlet oxygen (^1^O_2_) and superoxide anion (O_2_^−^) existing in the air. Thus, the reactions between the designed dyes and two kinds of reactive oxygen were analyzed by the computational and quantum chemistry method to estimate the photo-stability of the designed dyes.

As shown in Figure 2, the highest occupied molecular orbitals (HOMO) and lowest unoccupied molecular orbitals (LUMO) of the designed dyes were calculated by the Gaussian 09. The atoms were numbered in Appendix A. Comparing the HOMOs and LUMOs of the dyes, the different substituents and their positions on the pyridinium ring have varied influence, especially for the dyes of Z1 and Z4. As shown in Table 4 and Appendix A, the HOMO and LUMO orbitals of ^1^O_2_, O_2_^−^, ^3^O_2_, and Z1–Z6 were calculated based on hf/6-31 + g (d).

According to the basic principles of linear combination of atomic orbitals (LCAO)-molecular orbital theory (MO), HOMO of the dyes and LUMO of the reactive oxygen species should have similar symmetry, energy levels, as well as the maximum overlap in the orbitals. Obviously, the HOMO orbitals energy of the dyes is close to that of the LUMO orbital of ^1^O_2_, while the energy of HOMO orbital of O_2_^−^ is approximate to that of the LUMO orbitals of the dyes. Thus, the ^1^O_2_ LUMO orbitals could react with the HOMO orbitals of the dyes, while the O_2_^−^ HOMO orbitals could react with the LUMO orbitals of the dyes—causing photo-oxidation. The O_2_^−1^ could be more powerful than ^1^O_2_ in photo-oxidation process since the energy gap between the LUMO orbitals of the dyes and O_2_^−^ HOMO orbitals are lower than energy gap between the HOMO orbital of the dyes and ^1^O_2_ LUMO orbital (see Appendix A). The active positions of HOMO and LUMO orbitals of the dyes are summarized from Figure 2 and Appendix A, and listed in Table 5.

The dyes Z1–Z6 have similar active positions in HOMO and LUMO orbitals due to the similar chemical structures. The atomic orbital coefficient of the dye Z4 in HOMO orbital was higher than that of the Z1, while the energies of HOMO orbitals of the dyes Z2, Z3, Z5, and Z6 are close to each other and lower than that of Z1, indicating that the dye Z4 would be more active than other dyes. Also, the dyes Z2, Z3, Z5, and Z6 would be more stable than the dye Z1 during the photo-oxidation process in terms of the lower atomic orbital coefficients in HOMO orbitals. The LUMO orbitals of the dyes are more complicated since they have three active positions in LUMO orbitals. Three atomic orbital coefficients (green values) of the dyes Z2, Z5, and Z6 were lower than that of the dye Z1, while the other atomic orbital coefficients of the dyes are increased (red values) or decreased compared to the dye Z1. However, the atomic orbital coefficients of N45-C3 in the dye Z5 are lower than that of N46-C3 in dye Z1, and the atomic orbital coefficients of the N33-C16 in dye Z5 are higher than that of N34-C17 in dye Z1. The same situation could be found in dye Z2, meaning that dyes Z2, Z5, and Z1 have similar reactive LUMO orbitals. Based on the above analysis and Appendix A, dyes Z2 and Z5 would have better photo-stability than the dyes Z3, Z4, and Z6, but it is hard to determine whether dyes Z2 and Z5 have better photo-stability than dye Z1.

### 2.5. Synthesis and Applications of Dyes Z2 and Z5

Following the theoretical speculations on ideal fluorescent dyes for textile applications, dyes Z2 and Z5 were selected to prove the practicality of the theory. The prepared dyes Z2 and Z5 were confirmed by FTIR, UV-vis, and fluorescent spectra (Figure 3). As shown in Figure 3a, the FTIR spectrum of dye Z2 is similar to that of dye Z5 due to the similar chemical structures. The differences in the FTIR of dyes Z2 and Z5 are marked in different colors in Figure 3a. Obviously, there are characteristic absorption peaks (1743 cm^−1^, 1243 cm^−1^) of carboxyl group in Z5, while the 632 cm^−1^ is the absorption peak of S=O in sulfonic group of Z2. The absorption peaks of carbon double bonds (–C=C–) of dye Z2 and Z5 are about 1630 cm^−1^ and 1650 cm^−1^, respectively, while the absorption peaks of pyridine (Py) of dyes Z2 (1574 cm^−1^) and Z5 (1577 cm^−1^) are slightly different.

Ethanol solutions of Z2 and Z5 under UV and D65 light display different colors (Figure 3b). According to Figure 3c, the experimental maximum absorbance wavelengths of dyes Z2 and Z5 were around 520 nm and 490 nm, and the measured maximum emission wavelengths of them were about 622 nm and 612 nm, respectively. The results indicated that the measured maximum absorption and emission wavelengths are close to these estimated data, especially for the dye Z5 (see Appendix A). The synthesized dyes (Z2 and Z5, Appendix A) and commercial Rhodamine B (Appendix A) were used to dye PAN, cellulose, and nylon (2 g/piece, woven, Shanghai Textile Industry Institute of Technical Supervision, Figure 3d and Appendix A) following Appendix A in an X-5 DYEING machine (Foshan HUANGJU, China) with liquor ratio 50:1 at pH 4.5–5.0 by an acetic acid–sodium acetate buffer solution. The dye solutions were prepared with dyes (1% owf), sodium sulphate (3 g/L), and surfactant (0.5 g/L). The temperature was raised from room temperature to 100 °C at the rate of 1 °C/min after the fabrics were immersed into the dye solutions. After that, dyeing took place at this temperature and continued for a further 60 min; the dye solutions were then cooled to 70 °C at 1.25 °C/min, and the dyed fabrics were washed thoroughly in distilled water and dried in the open air [39,40]. The three blank fabrics have no obvious absorption and emission peaks under visible light, while the six dyed fabrics displayed obvious absorption and emission peaks under the visible light, indicating that dyes Z2 and Z5 can color PAN, nylon, and cotton fabrics. However, different to the dyed PAN and Nylon, the cotton fabrics dyed by Z2 and Z5 displayed small adsorption and emission intensity. Similar to the water solutions of dyes, the maximum absorption wavelengths of dye Z2 on PAN and Nylon were around 510 nm, and the maximum emission wavelengths of them were around 625 nm. The maximum absorption wavelengths of dye Z5 on PAN and Nylon were about 475 nm and 505 nm, while the maximum emission wavelengths of dye Z5 on PAN and nylon were around 610 nm and 615 nm, respectively.

As shown in Figure 2, the energy of HOMO orbitals of Z2 and Z5 were −0.36080 hartree and −0.35420 hartree, so the energy gap between the HOMO orbital of dye Z2 and ^1^O_2_ LUMO (0.01129 hartree) orbital was greater than the energy gap between the HOMO orbital of dye Z5 orbitals and ^1^O_2_ LUMO orbital, which means that dye Z5 was more reactive than dye Z2 in terms of HOMO orbitals; similarly but different to the HOMO orbital, the energy gap between the LUMO (−0.10124 hartree) orbital of dye Z5 and O_2_^−^ HOMO (−0.12516 hartree) orbital was smaller than the energy gap between the LUMO (−0.09660 hartree) orbitals of dye Z2 and O_2_^−^ HOMO orbital. Thus, dye Z2 would be more stable than dye Z5 when it reacts with reactive oxygen species. However, when compared with Z1, the energy of the HOMO and LUMO orbitals of dye Z2 and Z5 were lower than that of dye Z1, so it is hard to determine whether dyes Z2 and Z5 have better photo-stability than dye Z1. According to Figure 4, the maximum reflectivity of the Z1 dyed fabric lost about 20% of initial color intensity after 5 h light exposure. The Z2 dyed fabric only displayed about 16% loss in color intensity after the same duration of light exposure, while the Z5 dyed fabric showed about 24% loss in color intensity, meaning that dye Z2 had better photo-stability than dye Z1, but dye Z5 showed slightly lower photo-stability than dye Z1 in dyed PAN fabrics. The results are a little disappointing but will be addressed in future work.

## 3. Materials and Methods

### 3.1. Software and Calculation Methods

Geometry structures of the designed dye were optimized by using Gaussian 09 based on the b3lyp method with 6-31G(d) basis sets [41,42,43], and their charge distributions, enthalpies, and Gibbs free energies were estimated at b3lyp/6-31G(d) level [41,44,45,46], while the maximum absorption and emission wavelengths of them were predicted by using time-dependent density functional theory (TD-DFT) at the b3lyp/6-31G(d) level [47,48,49,50,51,52]. ECOSAR was used to predict aquatic toxicity of the chemicals by using the database based on the quantitative structure–activity relationship (QSAR) [53,54]. The Hansen distances of the dyes to water and fiber materials were calculated based on HSPiP 4.1.07 and used to estimate the different affinities between dyes, water, and fiber materials.

### 3.2. Design and Feasibility Analysis of Dyes

#### 3.2.1. Route Design of Hemicyanine Dyes

As shown in Figure 5, the hemciayanine dyes could be prepared by 2 steps; the chemical structures of raw materials, intermediates, and dyes are shown in Appendix A, while the geometry structures of the designed dyes are shown in Appendix A. Z1 (DYE-BD) could be easily synthesized in two steps following the references [15,16,17,18], and the route of the synthesis can serve as an example to illustrate the designed routes. The nitrogen center of pyridine features a basic lone pair of electrons; consequently, pyridine is a strong nucleophile. Thus, pyridine could react with 1-bromoethane to form pyridinium, while the second reaction is a typical Knoevenagel condensation reaction between the methyl group on the pyridinium and the carbonyl group in 4-diethylaminobenzaldehyde.

Although the Z1 in Figure 5 has been reported and investigated by researchers [15,16,17,18], the properties of other chemicals could be changed by different substituent groups, and the reactivity of intermediates would be changed as well. Thus, analysis and design of proper synthesis routes are important and necessary. The feasibilities of the routes were analyzed by using the computational methods.

#### 3.2.2. Analysis of Charge Density of Intermediates

The first step of the preparation route of designed dyes in Figure 5 is a formation of quaternary pyridinium salts on pyridine derivatives. Due to the existence of unconjugated lone pair electrons on N in pyridine rings, all six derivatives of pyridine can form the corresponding pyridinium salts easily (see Appendix A). The second step reaction, a nucleophilic reaction, is between the pyridinium salts and 4-diethylaminobenzaldehyde, while the reactive sites on the pyridinium salts are carbons bearing more electrons and more negative charges.

Figure 6 displays the charge distributions of the pyridinium intermediates. As a result of calculation, for the pyridinium salt (g) in Figure 6, the carbon, with an electron density of −0.391, in the methyl group should be the reactive site when it reacts with 4-diethylaminobenzaldehyde, while the methyl group in compound B1 is more reactive than methyl group in compound A1, which can be confirmed in the references [55,56,57,58,59,60,61]. Thus, the more negative charges of carbon in methyl group, the more reactive the methyl group will be. Sulfonic acid and carboxyl groups have a complex influence on the charge of the atoms in pyridinium salts. The carboxyl group (2 position) increases the electronegativity of the carbon in methyl group slightly, but the carboxyl (3 position) decreases the charge on the carbon of the methyl group. Thus, the methyl group of compounds B2, B3, B4, and B6 should have the similar reactivity to that of the compound B1, while the reactivity of methyl group in compound B5 should be lower than that of the compound B1. Thus, compounds B2, B3, B4, and B6 could react with 4-diethylaminobenzaldehyde, but the compound B5 may be difficult to or may not react with it.

#### 3.2.3. Analysis of Enthalpy and Gibbs Free Energy

The enthalpy of a thermodynamic system is defined in Equation (2) [62,63].
H = U + pV(2)

H is the enthalpy (SI unit: Joule); U is the internal energy (SI unit: Joule); p is pressure (SI unit: Pascal); V is volume (SI unit: m3).

The standard enthalpy of formation (Δ_f_H) of a compound is the change of enthalpy during the formation of 1 mole of the substance from its constituent elements in their standard states. The standard enthalpy change (Δ_r_H) of any reaction can be calculated from the standard enthalpies of formation of reactants and products using Hess’s law. It could be illustrated in Equation (3):(3)$$\Delta_{r}H=\sum \Delta_{f}H\left(\mathrm{products}\right)-\sum \Delta_{f}H\left(\mathrm{reactants}\right)$$

The Gibbs free energy is defined in Equation (4):G = U + pV − TS = H − TS(4)

T is the temperature (SI unit: Kelvin); S is the entropy (SI unit: Joule per kelvin).

The Gibbs free energy change (ΔrG) of any reaction can be calculated based on Equation (5):(5)$$\Delta_{r}G=\sum \Delta_{f}G\left(\mathrm{products}\right)-\sum \Delta_{f}G\left(\mathrm{reactants}\right)$$

According to the first step reaction between derivatives of pyridine and 1-bromoethane (Figure 5a), the reaction could be illustrated as A + D→B, so the ΔrH could be calculated by Δ_r_H = Δ_f_H(B) − Δ_f_H(A) − Δ_f_H(D), while the ΔG could be illustrated as Δ_r_G = Δ_f_G(B) − Δ_f_G(A) − Δ_f_G(D). The changes of enthalpy and Gibbs free energies in the first step reactions are calculated by Gaussian 09 in ethanol at 80 °C and shown in Table 6; similarly, the ΔrH values of the second reaction in Figure 5b could be illustrated as Δ_r_H = Δ_f_H(Z) + Δ_f_H(F) − Δ_f_H(B) − Δ_f_H(E), while the ΔG values in Table 7 could be calculated by Δ_r_G = Δ_f_G(Z) + Δ_f_G(F) − Δ_f_G(B) − Δ_f_G(E). Meanwhile, the bromide ion is not involved in the second step reactions. It seems all designed fluorescent dyes could be produced according to the thermodynamic analysis.

### 3.3. Sample Synthesis Route

Preparation of Z2: 4-Methylpyridine-3-sulfonicacid (A2) (1.7319 g, 0.01 mol) and sodium bicarbonate (NaHCO_3_, 0.84 g, 0.01 mol) were dissolved in a mixture solution (ethanol/water = 2:1) and stirred at room temperature for several minutes. Then, 1-bromoethane (1.19 g, 0.011 mol) was added into the mixture solution, and the mixture was refluxed under 80 °C for 6 h. Afterward, 1.77 g of 4-diethylaminobenzaldehyde (0.01 mol) and several drops of pyridine were added into the mixture; keeping the temperature for 6 h, a crude product was obtained after the solvent was removed by vacuum distillation.

Preparation of Z5: 4-Methylpyridine-3-carboxylic acid (A5) (1.3714 g, 0.01 mol) and sodium bicarbonate (NaHCO_3_, 0.84 g, 0.01 mol) were dissolved in a mixture solution (ethanol/water = 2:1) and stirred at room temperature for several minutes. Then, 1-bromoethane (1.19 g, 0.011 mol) was added into the mixture solution, and the mixture was refluxed under 80 °C for 6 h. Afterward, 1.77 g of 4-diethylaminobenzaldehyde (0.01 mol) and several drops of pyridine were added into the mixture; keeping the temperature for 6 h, a crude product was obtained after the solvent was removed by vacuum distillation.

## 4. Conclusions

After computational analyses of design, synthesis, fluorescent properties, photo-stability, as well as estimation of aquatic environmental toxicity of the fluorescent dyes, a dye molecule (Z2) was identified as possessing desired properties and potential environmental friendliness. Both dyes Z2 and Z5 were prepared and proven to possess ideal fluorescent properties. The Z2 dyed fabrics showed improved light stability, while the Z5 dyed ones shown reduced light stability. The structures of the dye Z2 present features of containing a sulfonic acid group, which could reduce toxicity of the chemicals. Both carboxyl and sulfonic acid groups can improve dyeing properties/affinity to five polymers. The different substituent groups and positions have different influences on the photo-stability of dyes.

## Figures and Tables

**Figure 1 ijms-20-05971-f001:**
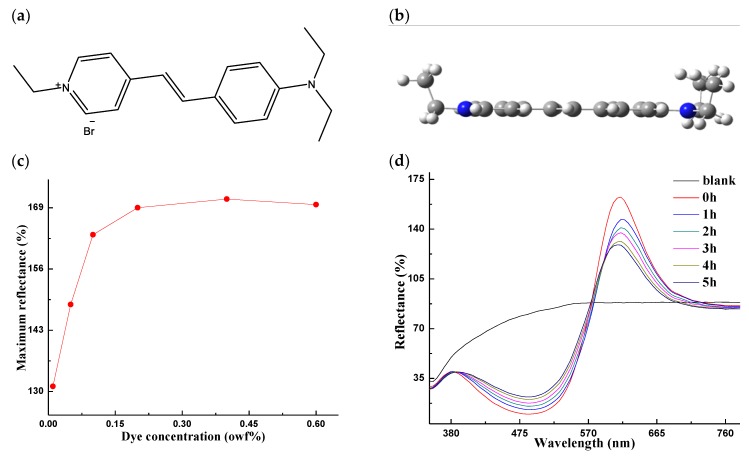
(**a**,**b**) Chemical and geometry structure of the hemicyanine–DYE BD; (**c**) maximum reflectance of dyed polyacrylonitrile (PAN) fabrics (% owf: 0.01; 0.05; 0.1; 0.2; 0.4; 0.6); (**d**) reflectivity of PAN fabrics dyed by DYE BD when it was exposed to light for different hours (0.4% owf).

**Figure 2 ijms-20-05971-f002:**
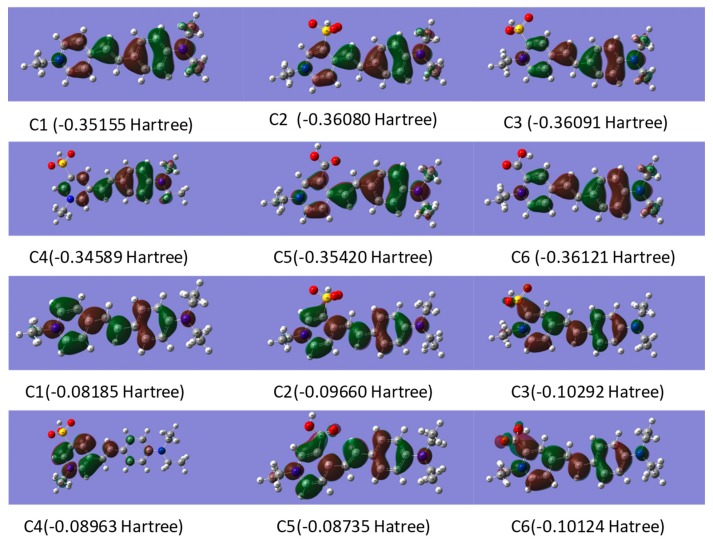
Highest occupied molecular orbitals (HOMO) (first six) and lowest unoccupied molecular orbitals (LUMO) (last six) orbitals of Z1–Z6.

**Figure 3 ijms-20-05971-f003:**
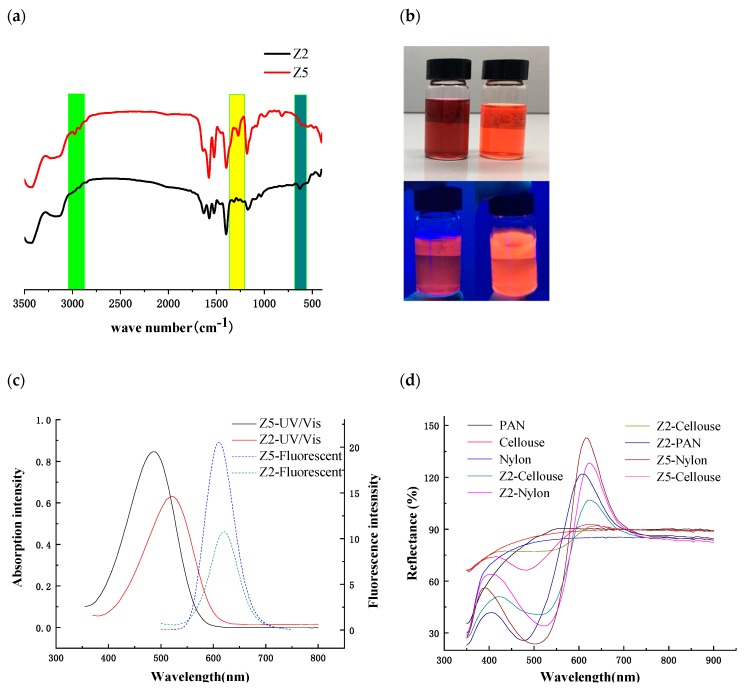
(**a**) FTIR spectral of dyes Z2 and Z5; (**b**) ethanol solutions of dyes Z2 (left) and Z5 (right) under D65 (upper) and UV (bottom) light; (**c**) absorption and emission spectra of dyes Z2 and Z5 in water; (**d**) reflectivity of different fabrics dyed by dyes Z2 and Z5.

**Figure 4 ijms-20-05971-f004:**
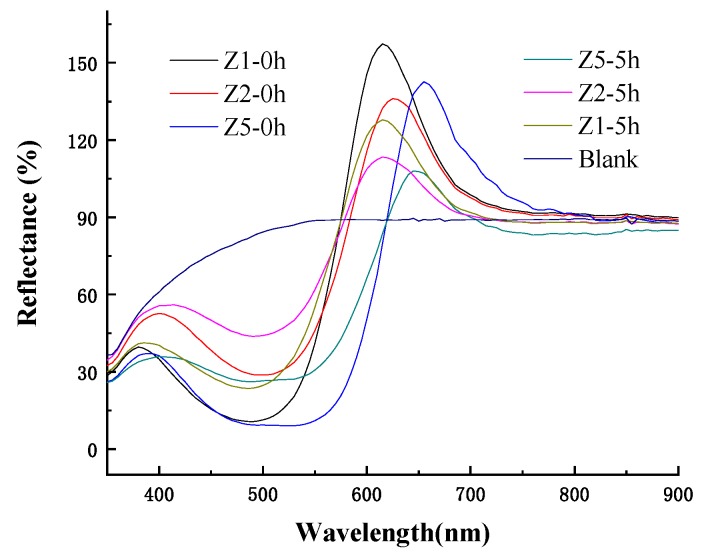
Light stability of the Z2 and Z5 dyed PAN fabrics exposed to the light for 0 and 5 h.

**Figure 5 ijms-20-05971-f005:**
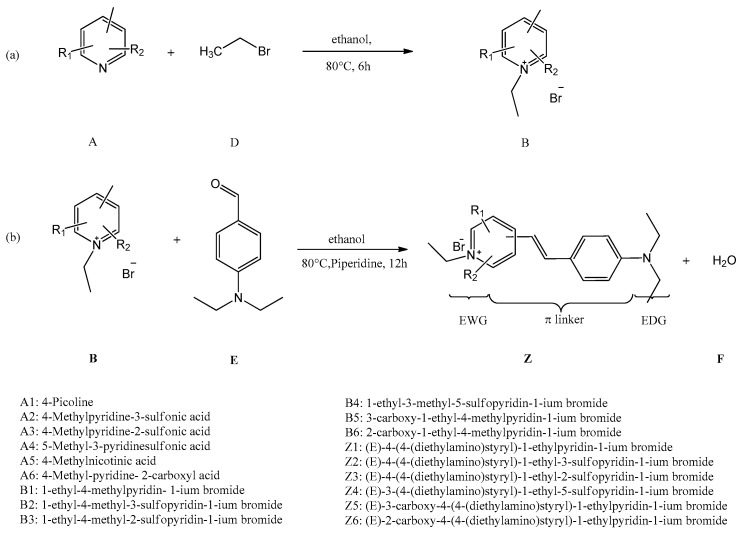
General method for prepare designed dyes. (**a**) formation of pyridinium; (**b**) preparation of designed dye.

**Figure 6 ijms-20-05971-f006:**
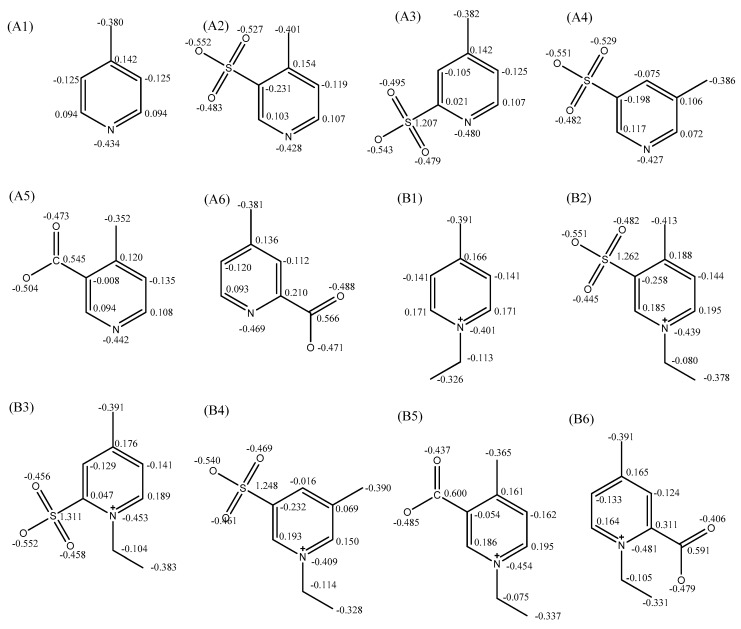
Electronegativity of raw materials and intermediates calculated by Gaussian 09. (A1) 4-Picoline; (A2) 4-Methylpyridine-3-sulfonic acid; (A3) 4-Methylpyridine-2-sulfonic acid; (A4) 5-Methyl-3-pyridinesulfonic acid; (A5) 4-Methylnicotinic acid; (A6) 4-Methyl-pyridine- 2-carboxyl acid; (B1) 1-ethyl-4-methylpyridin- 1-ium bromide; (B2) 1-ethyl-4-methyl-3-sulfopyridin-1-ium bromide; (B3) 1-ethyl-4-methyl-2-sulfopyridin-1-ium bromide; (B4) 1-ethyl-3-methyl-5-sulfopyridin-1-ium bromide; (B5) 3-carboxy-1-ethyl-4-methylpyridin-1-ium bromide; (B6) 2-carboxy-1-ethyl-4-methylpyridin-1-ium bromide.

**Table 1 ijms-20-05971-t001:** Estimated bio-toxicity of raw materials, intermediates, and designed dyes.

Item	Name	Fish 96h-LC_50_ (mg/L)	Daphnid 48h-LC_50_ (mg/L)	Green Algae 96h-EC_50_ (mg/L)
Raw materials	A1	292.3	156.33	90.92
	A2	3,727,904	1,489,483.88	259,458.53
	A3	3,727,904	1,489,483.88	259,458.53
	A4	3,727,904	1,489,483.88	259,458.53
	A5	11,169.25	5724.55	2792.02
	A6	91.92	5.36	8.56
Intermediates	B1	15,283,593	5,209,308.5	470,385.34
	B2	11,033,896,960	3,153,353,728	137,435,888
	B3	11,033,896,960	3,153,353,728	137,435,888
	B4	11,033,896,960	3,153,353,728	137,435,888
	B5	482,951,392	157,749,760	11,945,150
	B6	391,939.94	2527.1	61,436.25
Designed dyes	Z1	3094.48	1582.45	764.68
	Z2	25,924,188	9,904,030	1,433,391.12
	Z3	25,924,188	9,904,030	1,433,391.12
	Z4	25,924,188	9,904,030	1,433,391.12
	Z5	90,077.77	44,143.83	17,888.29
	Z6	588.73	27.57	57.77

**Table 2 ijms-20-05971-t002:** Fluorescence properties calculated by b3lyp/6-31G(d) for designed dyes in water.

Item	Max Absorption Wavelength			Max Emission Wavelength			Stokes (nm)
	^1^E (ev)	Wavelength (nm)	^2^f	E (ev)	Wavelength (nm)	f	
Z1	2.6105	474.94	1.4785	2.2869	542.15	1.5166	67.21
Z2	2.5576	484.77	1.4978	2.2167	559.32	1.5085	74.55
Z3	3.3454	370.61	0.1305	-	419.61	1.5554	49
Z4	2.6517	467.57	1.1095	2.4054	515.45	0.9673	47.88
Z5	2.4842	499.09	1.3926	2.0643	600.60	1.2270	101.51
Z6	2.3827	520.35	1.2081	1.7730	699.31	0.8955	178.96

^1^E: the energy of the light. ^2^f: oscillator strength.

**Table 3 ijms-20-05971-t003:** Hansen solubility parameter (HSP) values of the different dyes and Hansen distances to water (Ra1), cellobiose (Ra2), polyacrylonitrile (Ra3), poly (ethylene terephthalate) (Ra4), Nylon 6 (Ra5), and Nylon 66 (Ra6).

Dye	HSP (MPa1/2)			Ra1 (MPa1/2)	Ra2 (MPa1/2)	Ra3 (MPa1/2)	Ra4 (MPa1/2)	Ra5 (MPa1/2)	Ra6 (MPa1/2)
	δ_D_	δ_P_	δ_H_						
**Z1**	**18.4**	**3.0**	**1.7**	**43.02**	**23.69**	14.48	9.22	9.90	9.99
Z2	18.9	8.4	10.4	33.49	13.64	9.47	7.69	4.47	4.53
Z3	18.9	8.4	10.4	33.49	13.64	9.47	7.69	4.47	4.53
Z4	18.9	8.4	10.4	33.49	13.64	9.47	7.69	4.47	4.53
Z5	18.5	4.3	4.6	39.93	20.51	12.57	7.78	7.43	7.53
Z6	18.5	4.3	4.6	39.93	20.51	12.57	7.78	7.43	7.53
CI Disperse Yellow 11	20.2	5.3	8.7	36.49	16.64	12.53	8.27	7.08	7.16
H_2_O	15.5	16	42.3	0	20.26	36.09	39.78	36.51	36
Cellulose (cellobiose)	18.7	12.5	23.4	20.26	-	-	-	-	-
PAN	17.9	16.7	6.3	36.09	-	-	-	-	-
PET	19.6	11.7	3.6	39.78	-	-	-	-	-
Nylon 6	18.5	11.3	7.1	36.51	-	-	-	-	-
Nylon 66	18.5	11.4	7.1	36	-	-	-	-	-

**Table 4 ijms-20-05971-t004:** HOMO and LUMO orbital levels of ^3^O_2_, O_2_^−^, and ^1^O_2_ based on hf/6-31 + g(d).

Eigenvalues (Hartree)			^3^O_2_		^1^O_2_		O_2_^−^	
			HOMO	LUMO	HOMO	LUMO	HOMO	LUMO
			−0.56418	0.16997	−0.47706	0.01129	−0.12516	0.37411
Atomic orbital coefficients	O1	2S	0	0.12445	0	0	0	0.11664
		2PX	0	0	0	0.4592	0.49969	0
		2PY	0.54995	0	0.53402	0	0	0
		2PZ	0	−0.06966	0	0	0	−0.07467
	O2	2S	0	−0.12445	0	0	0	−0.11664
		2PX	0	0	0	−0.4592	−0.49969	0
		2PY	−0.54995	0	−0.53402	0	0	0
		2PZ	0	−0.06966	0	0	0	−0.07467

**Table 5 ijms-20-05971-t005:** HOMO/LUMO orbitals of active positions of dyes.

Item	HOMO 8		LUMO	
	Active Position	Atomic Orbital Coefficients	Active Position	Atomic Orbital Coefficients
Z1	N34-C17	0.27263, −0.13943	N46-C2	0.22269, −0.18852
			N46-C3	0.22269, −0.15797
			N34-C17	0.10212, −0.13939
Z2	N34-C17	0.26495, −0.1272	N45-C2	0.19891, −0.14321
			N45-C3	0.19891, −0.17116
			N34-C17	0.11526, −0.14978
Z3	N34-C17	0.26631, −0.13001	N44-C2	0.22645, −0.20734
			N44-C3	0.22645, −0.15098
			N34-C17	0.10354, −0.13363
Z4	N39-C18	0.28064, −0.14857	N43-C41	0.28690, −0.16810
			N43-C4	0.28690, −0.25597
			C2-C1	0.30619, −0.15157
Z5	N33-C16	0.26451, −0.13186	N45-C2	0.18462, −0.20217
			N45-C3	0.18462, −0.09251
			N33-C16	0.10986, −0.14634
Z6	N34-C17	0.26918, −0.1317	N44-C2	0.22579, −0.20415
			N44-C3	0.22579, −0.14513
			N34-C17	0.10063, −0.13091

**Table 6 ijms-20-05971-t006:** The enthalpy and Gibbs free energy values of chemicals in first step and changes of them in ethanol under 353K based on Gaussian 09.

Item	Δ_f_H (Hartree)	Δ_f_G (Hartree)	Item	Δ_f_H (Hartree)	Δ_f_G (Hartree)	Δ_r_H (Hartree)	ΔG (Hartree)
A1	−287.48332	−287.53033	B1	−2938.370434	−2938.438338	−0.022028	−0.002863
A2	−911.256057	−911.316383	B2	−3562.148829	−3562.223358	−0.027686	−0.00183
A3	−911.264174	−911.326257	B3	−3562.139176	−3562.218212	−0.009916	0.01319
A4	−911.258738	−911.321063	B4	−3562.13095	−3562.212119	−0.007126	0.014089
A5	−476.036178	−476.091818	B5	−3126.917758	−3126.99308	−0.016494	0.003883
A6	−476.039513	−476.096557	B6	−3126.905813	−3126.980969	−0.001214	0.020733
D	−2650.865086	−2650.905145	-	-	-	-	-

**Table 7 ijms-20-05971-t007:** The enthalpy and Gibbs free energy values of chemicals in second step and changes of them in ethanol under 353K based on Gaussian 09.

Item	Δ_f_H (^1^Hartree)	Δ_f_G (Hartree)	Item	Δ_f_H (Hartree)	Δ_f_G (Hartree)	ΔH (Hartree)	ΔG (Hartree)
B1	−366.48955	−366.545969	Z1	−848.007794	−848.103322	0.015431	0.019032
B2	−990.24633	−990.315512	Z2	−1471.772855	−1471.88174	0.00715	0.010157
B3	−990.238486	−990.308045	Z3	−1471.762939	−1471.871835	0.009222	0.012595
B4	−990.249985	−990.320103	Z4	−1471.761824	−1471.871864	0.021836	0.024624
B5	−555.034403	−555.097776	Z5	−1036.548901	−1036.654302	0.019177	0.019859
B6	−555.02708	−555.092909	Z6	−1036.547848	−1036.651849	0.012907	0.017445
E	−557.924048	−557.992916	F	−76.390373	−76.416531	-	-

^1^Hartree = 627.509 kcal mol^−1^ = 27.2116 eV.

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
