# Peer review of "Computer-Assisted Design of Environmentally Friendly and Light-Stable Fluorescent Dyes for Textile Applications"

_ijms, 2019, doi:10.3390/ijms20235971_

Round 1
Reviewer 1 Report
The paper is well written and prepared. The methodology is clear and I recommend to publish this paper in recent form.
Author Response
Thanks for you review.
Reviewer 2 Report
The paper is extremely interesting but is lucking to describe application methods to PAN cellulose and nylon.
How the application was made ?
It is critical and should be mentioned in the paper !!!
Author Response
How the application was made? It’s critical and should be mentioned in the paper!!!.
Answer: Thanks for your comment. The applications synthesic dyes and commercial dye rhodamine B on PAN, celloluse and Nylon are added in the supporting information and mentioned in the text, and the dyeing process of PET fabric is also added in the supporting information too.
The sentence “The dyes (Z2 and Z5) were employed in dyeing PAN, nylon and cotton fabrics (Figure 5d and Figure S7a) “ in the paper are changed as “The dyes (Z2 and Z5) were employed in dyeing PAN, nylon and cotton fabrics (Figure 5d and Figure S7a) following the traditional dyeing method of cationic dyes (Figure S10)”
Reviewer 3 Report
The paper describes a novel approach using computer assistance for evaluationof textile dyes. It has an environmental aspect and quite up to date. I only
suggest English editing by professional language editor.
can be accepted
Author Response
thanks for your review. The language was edited by an professional language eritor.
Round 2
Reviewer 2 Report
Still is not clear, plaese have a look on the following:
295 The dyes (Z2 and Z5) were employed in dyeing PAN, nylon and cotton fabrics (Figure 5d and Figure 296 S7a) following the traditional dyeing method of cationic dyes (Figure S10)Sorry but there is no traditional dyeing method using cationic dyes for all above substrates !!!!! if you have one please write it down !!! This is a serious omission in your paper if not mentioned how you performed the dyeing I can not give my acceptance !!!You have done an excellent work but the important part is missing !!!
Author Response
Thanks for the comment. The application methods of synthesized dyes and commercial rhodamine B on PAN, cotton and Nylon are revised and added in the manuscript, and the dyeing process of PET fabric is also added in the supporting information.
The following sentences were added in the manuscript. “The synthesized dyes (Z2 and Z5) and commercial Rhodamine B dye were used to dye PAN, cellulose and nylon (Figure 5d and Figure S7a) following the Figure S10 in an X-5 DYEING machine (Foshan HUANGJU, China), and the dye solutions were prepared with the required amount of each dye, sodium sulphate (3g/L) and a surfactant (0.5g/L). The pH of the dye bath was maintained at 4.5-5.0 by an acetic acid-sodium acetate buffer solution. The liquor-to-goods ratio was kept at 50:1. After immersing the fabrics into the dye solutions at room temperature, the temperature was increased to 100℃ at the rate of 1℃/min and maintained at the temperature for 60 minutes. Then, the dye solution was cooled to 70℃ at 1.25℃/min. At the end of the dyeing process, the dyed fabric was rinsed thoroughly in distilled water and allowed to dry in the open air.”